Increased level of FAM19A5 is associated with cerebral small vessel disease and leads to a better outcome

Hao Zhongnan
Yang Shaonan
Yin Ruihua
Wei Jin
Wang Yuan
Pan Xudong xudongpan887@126.com
Ma Aijun h19960126@163.com
The Affiliated Hospital of Qingdao University , Qingdao , China
Gould Gwyn
Electronic publication date: 2022 Mar 8
Publication date: 2022
Volume: 10
Electronic Location ID: e13101
Received 2022 Jan 13; Accepted 2022 Feb 21
Copyright: ©2022 Hao et al.
Copyright year: 2022
Copyright holder: Hao et al.
License: This is an open access article distributed under the terms of the Creative Commons Attribution License, which permits unrestricted use, distribution, reproduction and adaptation in any medium and for any purpose provided that it is properly attributed. For attribution, the original author(s), title, publication source (PeerJ) and either DOI or URL of the article must be cited.
License URL: https://creativecommons.org/licenses/by/4.0/

Keywords: FAM19A5, Cerebral small vessel disease, Recent small subcortical infarct, White matter hyperintensity, Enlarged perivascular space

Funding: The National Natural Science Foundation 81971111 82001251 This study was funded by the National Natural Science Foundation (81971111) and the National Natural Science Foundation (82001251). The funders had no role in study design, data collection and analysis, decision to publish, or preparation of the manuscript.

==============================
Objective

FAM19A5 plays an essential role in the development and acute or chronic inflammation of the central nervous system. The present study aimed to explore the association between FAM19A5 and cerebral small vessel disease (cSVD).

Methods

A total of 344 recent small subcortical infarct (RSSI) patients and 265 healthy controls were included in this study. The difference in the FAM19A5 level between the two groups was compared and the correlation between FAM19A5 and cerebral infarction volume was analyzed. Also, the association between FAM19A5 and the total magnetic resonance imaging (MRI) burden with its imaging characteristics was explored. Moreover, the correspondence of FAM19A5 with the outcome was assessed via Δ National Institutes of Health Stroke Scale score (NIHSS) and the percentage of NIHSS improvement.

Results

FAM19A5 was highly expressed in the RSSI group (P = 0.023), showing a positive correlation with cerebral infarction volume (P < 0.01). It was positively correlated with total MRI cSVD burden (P < 0.001) and reflected the severity of white matter hyperintensity (WMH) (P < 0.01) and enlarged perivascular space (EPVS) (P < 0.01), but did not show any association with cerebral microbleed (CMB) and lacune. Moreover, FAM19A5 suggested a larger Δ NIHSS (P = 0.021) and NIHSS improvement percentage (P = 0.007).

Conclusion

Serum FAM19A5 was increased in RSSI and positively correlated with the infarct volume. It also reflects the total MRI burden of cSVD, of which the imaging characteristics are positively correlated with WMH and EPVS. In addition, higher FAM19A5 levels reflect better outcomes in RSSI patients.

Introduction

Recent small subcortical infarction (RSSI), formerly known as lacunar infarct, accounts for 25% of all acute ischemic strokes. It results from the occlusion of a minor, single perforating brain artery, supplying subcortical areas in the deep white matter, basal ganglia (BG), thalamus, or pons (Cannistraro et al., 2019; Gattringer et al., 2015). Typically, the core mechanism of cSVD-related brain acute injury is considered to be ischemia, which effectuates through arteriolar stenosis or occlusion, irrespective of structural or functional (such as vasospasm, impaired self-regulation, or hypotension) (Wardlaw, Smith & Dichgans, 2013a). RSSI is one of the critical MRI characteristics of cerebral small vessel disease (cSVD). Lacune, white matter hyperintensity (WMH), enlarged perivascular space (EPVS), and cerebral microbleed (CMB) are also components of cSVD imaging characteristics (Wardlaw et al., 2013b). The main symptoms of cSVD are stroke (subcortical infarction and cerebral hemorrhage), cognitive and affective disorders, gait disorder, and overall function decline. The pathogenesis of cSVD is complicated; various MRI imaging characteristics and different parts of lesion sites show unique pathogenesis that is independent and interrelated (Blanco, Müller & Spence, 2017; Satizabal et al., 2012; Wardlaw, Smith & Dichgans, 2013a).

FAM19A5, also known as TAFA-5, belongs to the family with sequence similarity 19 comprising five highly homologous genes (FAM19A1-5) (Tom Tang et al., 2004). Using X-gal staining in FAM19A5-LacZ knock-in mice, a previous found that FAM19A5 participates in the nervous system development from an early stage, expresses in neurons, oligodendrocyte precursor cells (OPCs), astrocytes, and microglia. Also, using the traumatic brain injury (TBI) model, FAM19A5 is suggested to be involved in the OPC-mediated repair process as an acute inflammatory factor after pathological brain injury (Shahapal et al., 2019). The participation of FAM19A5 in the TNF-α-induced hypothalamus inflammatory response also confirmed the role of FAM19A5 in acute inflammation (Kang et al., 2020). Moreover, as a bridge connecting chronic inflammation, reactive gliosis, and degenerative diseases in the brain, FAM19A5 is overexpressed in severe depressive disorder, Alzheimer’s, Parkinson’s disease, and vascular dementia (Han et al., 2020; Li et al., 2020; Li et al., 2021; Mez et al., 2017).

Several studies demonstrated that inflammation plays a crucial role in the pathogenesis of cSVD (Blanco, Müller & Spence, 2017; Kwee & Kwee, 2007; Satizabal et al., 2012; Wardlaw, Smith & Dichgans, 2013a); whether FAM19A5 affects the pathogenesis of cSVD as an inflammatory factor has not yet been published. Based on the association among laboratory, imaging, and clinical assessments, the present study aimed to explore the correlation between FAM19A5 and cSVD.

Methods

Subjects

Cerebral infarction patients and healthy individuals admitted to the Institute of Cerebrovascular Diseases Cooperative Medical Institutions from May 2018 to June 2020 were continuously enrolled in this study. RSSI is defined as a recent deep small infarction caused by a single perforating artery occlusion, leading to ischemic necrosis of the brain tissue in its blood supply area. The inclusion criteria were as follows: RSSI patients showing typical cerebral infarction symptoms and physical signs with ischemic focus <20 mm in brain MRI. The exclusion criteria were as follows: patients with carotid or intracranial artery stenosis >50%, cardiogenic stroke, previous cardiocerebrovascular disease, severe liver or kidney function injury, and inflammation-related diseases, such as rheumatic immune disease and tumor. This study was conducted following the Declaration of Helsinki, was approved by the Ethics Review Board of the Affiliated Hospital of Qingdao University (QYFY WZLL 26787), and informed consent was taken from all individual participants.

Clinical analysis of subjects

Data, including age, sex, vital signs (blood pressure and pulse), history (hypertension and diabetes), personal history (smoking and drinking history), and basic laboratory examinations (glucose, triglyceride, total cholesterol, low-density lipoprotein, high-density lipoprotein, white blood cell count, and uric acid), were collected from medical records of all the subjects at the time of registration. All the healthy controls underwent electrocardiography (ECG) and brain imaging examinations (head computed tomography (CT) scan or head MR scan), while the RSSI patients received the following examinations: cerebral MRI, carotid artery examination, and cerebrovascular examination (arterial CTA or MRA), cardiac ultrasound, and 24-h Holter electrocardiogram. A total of 146 RSSI patients completed the head susceptibility-weighted imaging (SWI) examination. RSSI patients underwent laboratory examinations with respect to C-reactive protein, high sensitivity C-reactive protein, and homocysteine. We also evaluated the patients’ NIHSS score at admission and 14 days and calculated ΔNIHSS (NIHSS at admission–NIHSS at 14 days) and percentage of NIHSS improvement ((NIHSS at admission–NIHSS at 14 days)/NIHSS at admission). Also, the total MRI burden of patients in the RSSI group was estimated.

Serum analysis for FAM19A5 levels

Fasting blood was withdrawn within 24 h of admission, and the serum was collected by centrifugation at 1,000× g for 15 min and stored at −80 °C. The serum FAM19A5 level of all patients was determined by a double sandwich enzyme-linked immunosorbent assay kit (Jonln Biology, Wuhan, China), with inter-assay CV% < 10%, and intra-assay CV% < 15%, using 50 µL of each sample with no sample dilution. We strictly follow the operating instructions: the standard, specimen, and HRP-conjugate detection antibody were added to the microwells pre-coated with the FAM19A5 capture antibody in sequence. After incubation and washing thoroughly, the color was developed by TMB, and the optical density was measured at 450 nm on a microplate reader (Molecular Devices, San Jose, CA, USA).

Radiological assessment

All RSSI patients were examined using 3.0T MRI, and the images were read and analyzed by two professional neurologists; any differences were resolved through discussion with a third analyst. The volume of cerebral infarction was calculated using the ABC/2 method (Sims et al., 2009). Each imaging characteristic of the cSVD (WMH, EPVS, lacune, CMB) was evaluated separately. The lacune of presumed vascular origin has defined as a round or ovoid, subcortical, fluid-filled cavity (signal similar to CSF) between 3 mm and 15 mm in diameter, consistent with a previous acute small subcortical infarct or hemorrhage in the territory of one perforating arteriole. EPVS has defined as fluid-filled spaces that follow the typical course of a vessel as it goes through grey or white matter. The spaces have a signal intensity similar to CSF on all sequences. According to Potter scale, we counted the number of enlarged perivascular spaces and divided them into 0, 1, 2, 3, and 4 grades (0, 1–10, 11–20, 21–40, ≥40, respectively) (Potter et al., 2015). WMH has been defined as a signal abnormality of variable size in the white matter that shows the following characteristics: hyperintensity on T2-weighted images (T2WI) such as fluid-attenuated inversion recovery without cavitation (signal different from CSF); WMH was divided into periventricular WMH (PVWMH) and deep WMH (DWMH) that were graded by Fazekas scale, respectively (Wardlaw et al., 2013b), Only RSSI patients with head SWI were evaluated for CMB, which was defined as a small focal round lesion (diameter <10 mm), showing a low signal on SWI. The CMB was classified into lobar CMB and deep CMB. The presence of lacune, the presence of CMB, the number of EPVS ≥11, Fazekas scale ≥2, and each parameter assessed as one score were included in the total MRI burden of cSVD.

Statistical analysis

The normality of quantitative data distribution was assessed using the Kolmogorov–Smirnov test. Continuous variables with normal distribution were described as mean ± standard deviation (SD), continuous variables with non-normal distribution were described as median (interquartile range (IQR)), and qualitative data were described as quantity (percentage). Multivariable logistic regression analysis compared the difference in FAM19A5 expression between the RSSI group and healthy controls, while multivariable linear regression analysis compared the correlation between FAM19A5 and the infarct volume. In order to verify the correlation between the imaging characteristics of total MRI burden and FAM19A5, we first determined the confounding factors by univariate logistic analysis, and those with P-value < 0.1 were included in multivariable logistic analysis for adjustment. Due to the high incidence of WMH and EPVS, the variables were classified into different grades that were analyzed by multivariate ordinal logistic regression. Next, lacune and CMB were analyzed by multivariate binary logistic regression with the classification into the presence and absence. Deep and lobar CMB and, as well as PVWMH and DWMH were independently analyzed by multivariate logistics. P < 0.05 was considered statistically significant. In the RSSI group, the correlation between FAM19A5 levels and clinical scales (NIHSS scale at NIHSS scale at admission, NIHSS scale at 14 days, ΔNIHSS, percentage of NIHSS improvement, and total MRI burden of cSVD) were assessed, in which Spearman’s test performed the comparisons between continuous variables. All statistical analyses were performed using IBM SPSS Statistics for Windows, version 26.0 (IBM Corporation, Armonk, NY, USA).

Results

Recruitment and baseline characteristics of subjects

A total of 609 subjects were selected in our study from 1784 RSSI patients and healthy individuals from May 2018 to June 2020 (Fig. 1). There were 1,342 acute infarction patients were included in the filter process; among them, 573 patients with severe large artery stenosis and occlusion were excluded, and 188 patients were excluded because of cardiac diseases or other diseases that may lead to cerebral infarctions. Finally, 344 RSSI patients were included in the current study, the mean age was 61.874 ± 11.359 years, and 211 (63.2%) patients were males. Also, 265 healthy controls were strictly filtered from 541 patients undergoing health checkups, the mean age was 61.411 ± 11.978 years, and 115 (43.4%) patients were males (Table 1).

Figure 1 Recruitment flowchart.

Table 1 Baseline characteristics of RSSI group and healthy controls.

	RSSI (n = 334)	Healthy control (n = 265)	
Age, years (SD)	61.874  ±  11.359	61.411 (11.978)	
Sex, male (n%)	211 (63.173)	115 (43.396)	
hypertension, n (n%)	207 (61.976)	131 (49.434)	
Diabetes, n (n%)	90 (27.245)	44 (16.604)	
Current smoking, n (n%)	121 (36.228)	33 (13.415)	
Pulse, beats/min (IQR)	68 (59–77)	69 (63–76)	
SBP, mmHg (IQR)	151 (137–167)	133 (120–149)	
DBP, mmHg (IQR)	84 (76–95)	76 (68–84)	
FAM19A5, ng/mL(SD)	0.682  ±  1.225	0.433  ±  0.778	
Notes.

SBP systolic blood pressure

DBP diastolic blood pressure

FAM19A5 family with sequence similarity 19 memberA5

FAM19A5 level and infarct volume of RSSI

The current study showed a positive correlation between FAM19A5 and recent small subcortical infarct. Mean ± standard deviation of serum FAM19A5 in the RSSI and healthy control groups was 681.894 ± 1225.014 and 432.582 ± 777.871 pg/mL, respectively (Fig. 2). After adjustment for age, sex, hypertension, diabetes, smoking, pulse, triglyceride, cholesterol, low-density lipoprotein, and glucose, our results showed that the expression of FAM19A5 was higher in the RSSI group than the control group (odds ratio (OR) (95% confidence interval (CI)) = 1.265 (1.032–1.549), P = 0.023). The linear regression analysis of infarct persons showed the correlation between infarct volume and FAM19A5 (OR (95% CI) = 1.587 (1.324–1.901), P < 0.001) (Table 2).

Figure 2 FAM19A5 levels of RSSI group & healthy control.

Serum level of FAM19A5 (ng/mL) in the RSSI group (mean ± SD = 0.682 ± 1.225) is higher than in healthy control (mean ± SD = 0.433 ± 0.778), P-value = 0.023.

FAM19A5 and imaging characteristics of total MRI burden

Serum FAM19A5 levels in imaging characteristics of cSVD total MRI burden are depicted in Fig. 3. A positive correlation between the total MRI burden of cSVD and FAM19A5 levels was established (P = 0.041) and assessed by univariate and multivariable regression analysis. (Table 3) Age, sex, hypertension, diabetes, smoking, drinking, pulse, blood pressure, volume of cerebral infarction, glucose, C-reactive protein, triglyceride, total cholesterol, low-density lipoprotein, high-density lipoprotein, white blood cell count, uric acid, and homocysteine were included in the adjustment process.

Table 2 Correlation between FAM19A5 and RSSI with its infarct volume.

	Unadjusted	Adjusteda	
	OR (95% CI)	P-value	OR (95% CI)	P-value	
Model 1b					
RSSI group vs. healthy controls	1.029 (1.077–1.559)	0.006	1.265 (1.032–1.549)	0.023	
Model 2c					
FAM19A5 and infarct volume	1.597 (1.339–1.906)	<0.001	1.587 (1.324–1.901)	<0.001	
Notes.

a Adjusted for age, sex, hypertension, diabetes, current smoking, pulse, triglyceride, cholesterol, lowdensity lipoprotein, glucose.

b Binary logistic regression analysis of FAM19A5 level between RSSI group and heathy controls.

c Linear regression analysis of FAM19A5 level and infarct volume of RSSI patients. R2 = 0.094. Bold indicates statistically significant P-values (P < 0.05).

Figure 3 Level of FAM19A5 and imaging characteristics of total MRI burden.

FAM19A5 level in persons with Fazekas scale 0, 1, 2, and 3: 0.144 ± 0.056, 0.415 ± 0.814, 0.745 ± 1.211, and 1.761 ± 2.001, respectively. (n = 4, 182, 116, and 42, respectively. P < 0.001). FAM19A5 level in persons with Potter scale 1, 2, 3, and 4: 0.276 ± 0.564, 0.751 ± 1.392, 1.065 ± 1.423, and 1.0162 ± 1.388, respectively. (n = 114, 122, 84, and 24, respectively. P < 0.001). FAM19A5 level in persons without or with Lacune: 0.777 ± 1.372 and 0.635 ± 1.134, respectively. (n = 125 and 219, P = 0.844). FAM19A5 level in persons without or with CMB: 0.736 ± 1.395 and 0.614 ± 0.870, respectively. (n = 95 and 51, P = 0.479). CMB, cerebral microbleed; EPVS, enlarged perivascular space; WMH, white matter hyperintensity.Bold indicates statistically significant P-values (P < 0.05). FAM19A5 level was described as mean ± standard deviation.

After adjustment for age, hypertension, pulse, systolic blood pressure, total cholesterol, low-density lipoprotein, C-reactive protein, and high sensitivity C-reactive protein, multivariate ordinal logistic regression analysis showed a positive correlation between FAM9A5 and white matter hypertension on Fazekas scale (OR (95% CI) =1.842 (1.459–2.326), P < 0.001). (Table 3) Our further analysis showed that FAM19A5 was correlated with both periventricular white matter hypertensity (OR (95% CI) = 1.560 (1.311–1.855), P < 0.001) and deep white matter hypertensity (OR (95% CI) = 1.557 (1.268–1.910), P < 0.001).

After adjustment for age, sex, hypertension, smoking, total cholesterol, and low-density lipoprotein, multivariate ordinal logistic regression analysis showed a positive correlation between FAM9A5 and EPVS Potter scale (OR (95% CI) = 1.434 (1.211–1.698), P < 0.001).

The current study did not identify any significant correlation between FAM9A5 and the presence or absence of lacunae (OR (95% CI) = 0.980 (0.801–1.199), P = 0.844) and CMB (OR (95% CI) = 0.895 (0.659–1.216), P = 0.479). We also found that deep microhemorrhage and cortical microhemorrhage were not related to the FAM19A5 level.

Correlation between FAM19A5 and outcomes

The mean ± SD of NIHSS at admission, NIHSS at 14 days, and ΔNIHSS was 3.22 ± 2.88, 2.52 ± 2.66, and 0.69 ± 1.27, respectively, and the mean ± SD of the percentage of NIHSS improvement was 25.4 ± 48.5. Spearman’s correlation analysis demonstrated that the FAM19A5 level was not significantly correlated with NIHSS at admission and 14 days after admission. However, it was positively correlated with ΔNIHSS (P = 0.031) and the percentage of NIHSS improvement (P = 0.011) (Table 4).

Table 3 Correlation between FAM19A5 and characteristics of cSVD total MRI burden.

	Unadjusted	Adjusted	
	OR (95% CI)	P-value	OR (95% CI)	P-value	
WMHa					
Fazekas scale	1.697 (1.408–2.044)	<0.001	1.842 (1.459–2.326)	<0.001	
EPVSb					
Potter scale	1.430 (1.213–1.687)	<0.001	1.434 (1.211–1.698)	<0.001	
Lacunec					
Presence or absence	0.913 (0.765–1.89)	0.311	0.980 (0.801–1.199)	0.844	
CMBd					
Presence or absence	0.917 (0.683–1.232)	0.567	0.895 (0.659–1.216)	0.479	
cSVDe					
Total MRI burden	1.183 (1.011-1.384)	0.036	1.251(1.010-1.550)	0.041	
Notes.

WMH, EPVS, and total MRI burden underwent ordinal logistic regression, Lacune and CMB were assessed by binary logistic regression.

a Adjusted for age, hypertension, pulse, SBP, TC, LDL, CRP, and hsCRP.

b Adjusted for age, sex, hypertension, smoking, TC, and LDL.

c Adjusted for age, hypertension, SBP, and Hcy.

d Adjusted for glucose, TC, and UA.

e Adjusted for age, SBP, glucose, TC, LDL, CRP, and Hcy.

SBP systolic blood pressure

TC total cholesterol

LDL low-density lipoprotein

Hcy homocysteine

CI volume cerebral infarction volume

CRP C-reactive protein

hsCRP high sensitivity C-reactive protein

Bold indicates statistically significant P-values (P < 0.05).

Table 4 Association of FAM19A5 levels with clinical scales of patients.

	RSSI (n = 334)	P-value	
NIHSS scale at admission	3.2173 ± 2.881	0.869	
NIHSS scale at 14 days	2.524 ± 2.657	0.109	
ΔNIHSS	0.693 ± 1.269	0.031	
Percentage of NIHSS improvement, %	25.361 ± 48.454	0.011	
Notes.

NIHSS National Institutes of Health stroke scale

Bold indicates statistically significant P-values (P < 0.05).

Discussion

The results showed that FAM19A5 increased in the RSSI group, showing a positive correlation with cerebral infarction volume. The multivariate logistic analysis showed that the level of FAM19A5 was positively correlated with the total MRI burden of cSVD, which might be effectuated through the correlation with WMH and EPVS. Moreover, FAM19A5 suggested a better outcome for RSSI patients. By combining the laboratory biomarker with clinical and imaging findings, we verified the association of FAM19A5 with the severity of cSVD and its prognosis, which might guide the treatment of cerebral small vessel diseases.

Reportedly, FAM19A5 participates in nervous system development at an early stage. It is expressed in neurons, oligodendrocyte precursor cells (OPCs), astrocytes, and microglia and is involved in the OPC-mediated repair process as an acute inflammatory factor after pathological brain injury (Shahapal et al., 2019). Previous studies have shown that FAM19A5 increases the level of pro-inflammatory cytokines and participates in the TNF-α-induced inflammatory response in the hypothalamus, which also confirms the role of FAM19A5 in acute inflammation (Kang et al., 2020). A clinical study demonstrated that FAM19A5 was increased in neuromyelitis optica spectrum disorders, the reason may be that after central nervous system (CNS) injury, FAM19A5 is secreted by reactive astrocytes, which triggers excessive proliferation of neuroglial cells and participates in the proliferation of reactive gliosis (Lee et al., 2019). Not only acute injury, but also FAM19A5 is involved in the pathophysiological process of chronic inflammation. As a bridge connecting chronic inflammation, reactive gliosis, and degenerative diseases in the brain, FAM19A5 is overexpressed in severe depressive disorder and negatively correlated with cortical thickness. It is also highly expressed in Alzheimer’s in genetic studies and vascular dementia (Han et al., 2020; Li et al., 2020; Mez et al., 2017).

Microglia and astrocytes are activated in acute infarction, leading to high expression of inflammatory factors via damage-associated molecular pattern (DAMPs) and “inflammatory waterfall” however, the correlation between FAM19A5 and RSSI has not yet been realized (Chen & Nuñez, 2010). To the best of our knowledge, we assessed the expression of FAM19A5 in cSVD and found that the expression level of FAM19A5 in RSSI patients was high and correlated with the infarct volume of RSSI. Based on the previous studies, we speculated that the reasons for the increased expression of FAM19A5 after cerebral infarction might be as follows: FAM19A5 is secreted by reactive glial cells and is involved in OPC-mediated repair process, triggers glial cell proliferation, and glial scar formation. This phenomenon is confirmed by the correlation between FAM19A5 and cerebral infarction volume: a large volume of brain injury indicates severe nerve damage and acute inflammation. Also, higher FAM19A5 predicted a larger ΔNIHSS and percentage of NIHSS improvement, suggesting that FAM19A5 contributes to the repair process of the nervous system via OPC-mediated gliosis and nerve repair process, thereby improving the stroke prognosis.

The mechanism of cerebral small vessel disease is complex, and the pathogenesis varies in imaging characteristics of cSVD and in different areas of the brain (Blanco, Müller & Spence, 2017). Deep CMB is closely related to hypertension, and lobar CMB might be caused by cerebral vascular amyloidosis (Ding et al., 2015; Poels et al., 2010; Wardlaw, Smith & Dichgans, 2013a). Several studies have shown that high signal intensity in white matter is correlated with the overexpression of various inflammatory factors (Satizabal et al., 2012). The disturbance of cerebrospinal fluid circulation in perivascular space or the imbalance of aquaporin in astrocytes might be one of the causes of EPVS. In addition, histopathological studies have shown that PVS is consistent with the location of inflammatory infiltration in EPVS (Kwee & Kwee, 2007). Lacune is usually considered to be transformed from other lesions (Wardlaw, Smith & Dichgans, 2013a; Wardlaw, Smith & Dichgans, 2019). Our correlation analysis results suggested that higher FAM19A5 levels reflect a severe total MRI burden, and multivariate logistic analysis showed that FAM19A5 reflects the severity of WMH and EPVS. Comprehensively, we speculated that the correlation between FAM19A5 and total MRI burden could be attributed to the effects on WMH and EPVS, which might be due to the involvement of FAM19A5 in chronic inflammation in the brain. Since WMH and EPVS are related to chronic and persistent inflammatory responses, the expression and activity of many inflammatory markers are elevated (Blanco, Müller & Spence, 2017; Kwee & Kwee, 2007; Satizabal et al., 2012). Herein, we did not find any correlation between deep or lobar CMB and FAM19A5. Comprehensively, the reason may be that CMBs are related to hypertension or amyloidosis, reflecting the deposition of hemosiderin in macrophages and failing to reflect the persistent inflammatory response of the brain (Janaway et al., 2014). In addition, lacunae are usually thought to be transformed from other lesions, and it is complex to trace encephalomalacia to its origins. Therefore, the etiology and mechanism are complex and lack solid evidence. Taken together, FAM19A5 is used as a chronic inflammatory factor that participates in the progression of cSVD and reflects its severity.

Nevertheless, the present study has several limitations. Firstly, because of our strict screening of patients with cerebral infarction, the study included only a small number of cases, and the various examination strategies of subcenters refer to insufficient tests in SWI examinations, thereby limiting the CMB research. Secondly, our subjects are homogeneous, and most of them are Han people in northern China. This problem could be solved by increasing the number of patients in the future. Thirdly, severe large artery stenosis was defined as stenosis >50% of a carotid or intracranial artery according to TOAST, it is the most broadly used but may leading to the insufficient exclusion of those with a high carotid plaque burden. Other classification systems like SPARKLE and CISS is meaningful in solving this problem in the fulture.(Bogiatzi et al., 2014; Zhang et al., 2019) Finally, due to the limitations of the cross-sectional analysis, we are unable to study causality. Thus, additional studies and dynamic observation are needed to solve this problem in the future.

In summary, FAM19A5 reflects the severity of RSSI and predicts a promising prognosis as an acute inflammatory factor. In the imaging characteristics of cSVD, FAM19A5 can reflect the severity of WMH and EPVS. That might guide the diagnosis and treatment of cerebrovascular diseases.

Supplemental Information

Supplemental Information 1 Raw data

The correlation between CSVD and FAM19A5.

Click here for additional data file.

Supplemental Information 2 OD of ELISA

Click here for additional data file.

Supplemental Information 3 OD of elisa

Click here for additional data file.

Supplemental Information 4 Operating instrution of FAM19A5 ElISA

Click here for additional data file.

We gratefully acknowledge the assistance of the Institute of Cerebrovascular Diseases Cooperative Medical Institutions for clinical support.

Additional Information and Declarations

Competing Interests

Author Contributions

Human Ethics

Data Availability

The authors declare there are no competing interests.

Zhongnan Hao conceived and designed the experiments, performed the experiments, analyzed the data, prepared figures and/or tables, authored or reviewed drafts of the paper, and approved the final draft.

Shaonan Yang, Ruihua Yin and Yuan Wang analyzed the data, prepared figures and/or tables, authored or reviewed drafts of the paper, and approved the final draft.

Jin Wei performed the experiments, prepared figures and/or tables, authored or reviewed drafts of the paper, and approved the final draft.

Xudong Pan and Aijun Ma conceived and designed the experiments, authored or reviewed drafts of the paper, and approved the final draft.

The following information was supplied relating to ethical approvals (i.e., approving body and any reference numbers):

The Ethics Review Board of the Affiliated Hospital of Qingdao University

The following information was supplied regarding data availability:

The raw measurements are available in the Supplementary File.

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
