# Peer review of "Increased level of FAM19A5 is associated with cerebral small vessel disease and leads to a better outcome"

_PeerJ, doi:10.7717/peerj.13101_

## Round 0.1 · original submission · Major Revisions

Both reviewers of this work found it to be of potential interest. reviewer 1 has a series of largely stylistic and information points for you to address, but reviewer-2 has raised some concerns regarding the assays and the statistics. These require careful consideration and detailed responses.

Please make sure in all data sets, legends, tables etc. that the sample size is recorded, the stats are clearly described and p-values presented for any points that are described as significant. Biological and technical replicates must also be provided for all datasets.

You also need to carefully address the points about the assay kit; without this information the paper will be significantly weakened.

Please make these changes and provide a cover letter in which the response to each point is carefully outlined.

·

Basic reporting

No comment

Experimental design

No comment.

Validity of the findings

No comment

Additional comments

Most readers will not be familiar with FAM19A5, so it is important to say that it is an acute inflammatory factor after pathological brain injury.

You say you excluded patients with severe large artery stenosis. Do you mean you excluded patients categorized as having Large Artery Atherosclerosis according to a classification such as TOAST? In that case it would be defined as stenosis >50% of a carotid or intracranial artery, not severe stenosis. There are problems with misclassification of LAA as ESUS or Unknown in TOAST; both the SPARKLE[1] and CISS classifications include patients with a high carotid plaque burden as well as 50% stenosis; those classifications identify more LAA and have fewer Unknown cases compared to TOAST.[2] If you used TOAST you should mention it as a limitation.

You give a nice definition of lacunar infarction, but then you seem to include all white matter lesions in the WMH. There is a large blood pressure gradient in the brain; “with blood pressure in the brachial artery of 117/75 mm Hg, the pressure in small parietal arterioles would be only 59/38 mm Hg”.[3] This blood pressure gradient probably affects different forms of small vessel disease differently.[4] Small subcortical infarctions in the distal reaches of the cerebral vasculature where blood pressure is low are probably not true lacunar infarctions, which occur in the vascular centrencephalon,[5, 6] where blood pressure is high (Basal ganglia, thalamus, internal capsule, brainstem). This was illustrated in a study of symptomatic vs. asymptomatic WMH. [7] Can you say if FAM19A5 was related to the location of white matter lesions?

In table 2 you mention “heat controls” in the heading, and “health controls” in the table; probably you mean “healthy controls”

1. Bogiatzi C, Wannarong T, McLeod AI, Heisel M, Hackam D, Spence JD. SPARKLE (Subtypes of Ischaemic Stroke Classification System), incorporating measurement of carotid plaque burden: a new validated tool for the classification of ischemic stroke subtypes. Neuroepidemiology. 2014;42(4):243-51.
2. Zhang H, Li Z, Dai Y, Guo E, Zhang C, Wang Y. Ischaemic stroke etiological classification system: the agreement analysis of CISS, SPARKLE and TOAST. Stroke Vasc Neurol. 2019;4(3):123-8.
3. Blanco PJ, Muller LO, Spence JD. Blood pressure gradients in cerebral arteries: a clue to pathogenesis of cerebral small vessel disease. Stroke Vasc Neurol. 2017;2(3):108-17.
4. Spence JD. Blood Pressure Gradients in the Brain: Their Importance to Understanding Pathogenesis of Cerebral Small Vessel Disease. Brain Sci. 2019;9(2).
5. Hachinski VC, Norris JW. The vascular infrastructure. The acute stroke. Philadelphia: F.A. Davis; 1985. p. 27-40.
6. Soros P, Whitehead S, Spence JD, Hachinski V. Antihypertensive treatment can prevent stroke and cognitive decline. Nat Rev Neurol. 2013;9(3):174-8.
7. Valdes Hernandez Mdel C, Maconick LC, Munoz Maniega S, Wang X, Wiseman S, Armitage PA, et al. A comparison of location of acute symptomatic vs. 'silent' small vessel lesions. Int J Stroke. 2015;10(7):1044-50.

Reviewer 2 ·

Basic reporting

Authors investigated the possible association of serum FAM19A5 levels with cerebral small vesicle disease. Their findings sound interesting. However, authors may need to add more clear information regarding results description and method validation for FAM19A5 determination.

Line 168- 169: Authors described Serum FAM19A5 in the RSSI and healthy control groups was 681.894±1225.014 and 432.582±777.871 ng/mL, respectively (Figure 2). In this sentence, authors may need to change “ng/mL” to “pg/mL”. Further, reviewer found significantly high variations of the levels. Therefore, they need to inform us whether the deviations are “standard error SE” or “standard deviation, SD”. Furthermore, this description is not correlated with the results shown in the Figure 2 where error bars show only ~ 20% variation. Authors may need to explain this discrepancy.

The authors used the ELISA kit provided by Jonln Biology (Wuhan) for serum FAM19A5 level determination. However, authors did not provide data regarding method validation. Reviewer would like to suggest adding more description on the issue in the Method section. In particular, considering the significant variations of FAM19A5 levels across individuals, they need to provide the inter-assay and intra-assay CV in the method section at least. Standard curve and dilution factor can be provided as well.

For Figure 3. Authors need to provide sample size (numbers) for each group. Again, reviewer found discrepancies between the described results and error bar size in the figure.

In addition, P values need to be provided in both Figures 2 and 3.

Figure 2 legends: the same sentence is duplicated.

References: “Shahapal A et al” is duplicated. Accordingly, author may need to revise citations in the main text, lines 6 and, 216

Experimental design

No more comment.

Validity of the findings

No more comment.

Additional comments

No more comment.

---

## Round 0.2 · Minor Revisions

Please correct the legend to Figure 2 as indicated by the reviewer then I will recommend acceptance.

Reviewer 2 ·

Basic reporting

Authors well addressed reviewers concerns to improve the manuscript.
I found that authors may need to correct Figure 2 legends by adding unit of FAM19A5 level eg “ng/ml”

Experimental design

no more comments

Validity of the findings

no more comments

---

## Round 0.3 · accepted · Accept

Thanks for tidying up the remaining points. Congratulations on a nice study.